# A Review of Issues Related to Data Acquisition and Analysis in EEG/MEG Studies

**DOI:** 10.3390/brainsci7060058

**Published:** 2017-05-31

**Authors:** Aina Puce, Matti S. Hämäläinen

**Affiliations:** 1Psychological & Brain Sciences, Indiana University, 1101 East 10th St, Bloomington, IN 47405, USA; 2Athinoula A. Martinos Center for Biomedical Imaging, Massachusetts General Hospital and Harvard Medical School, Charlestown, MA 02129, USA; msh@nmr.mgh.harvard.edu

**Keywords:** EEG, MEG, artifacts, data acquisition, reference electrode, data analysis, sensor space, source space

## Abstract

Electroencephalography (EEG) and magnetoencephalography (MEG) are non-invasive electrophysiological methods, which record electric potentials and magnetic fields due to electric currents in synchronously-active neurons. With MEG being more sensitive to neural activity from tangential currents and EEG being able to detect both radial and tangential sources, the two methods are complementary. Over the years, neurophysiological studies have changed considerably: high-density recordings are becoming de rigueur; there is interest in both spontaneous and evoked activity; and sophisticated artifact detection and removal methods are available. Improved head models for source estimation have also increased the precision of the current estimates, particularly for EEG and combined EEG/MEG. Because of their complementarity, more investigators are beginning to perform simultaneous EEG/MEG studies to gain more complete information about neural activity. Given the increase in methodological complexity in EEG/MEG, it is important to gather data that are of high quality and that are as artifact free as possible. Here, we discuss some issues in data acquisition and analysis of EEG and MEG data. Practical considerations for different types of EEG and MEG studies are also discussed.

## 1. Introduction: The Evolution of EEG/MEG Studies

EEG and MEG are excellent complementary methods that offer a non-invasive way to study (sub)millisecond brain dynamics. Many EEG/MEG studies in the latter part of the 20th century focused on studying evoked activity, i.e., activity that is time-locked to an incoming stimulus or an executed motor action. The bulk of EEG/MEG literature was thus devoted to the study of event-related potentials (ERPs) and event-related fields (ERFs). In research studies, spontaneously occurring activity was typically treated as noise, to be eliminated by averaging over multiple presentations of a stimulus. EEG has been used as a diagnostic tool almost since Berger’s first report in the late 1920s, and therefore, there is a rich clinical literature describing abnormal brain rhythms and evoked responses in various neurological and neuropsychiatric disorders.

EEG and MEG have now been available for approximately 80 and 40 years, respectively. Over the years, a number of consensus-based documents of best practices have been published for EEG research studies [1,2], as well as for clinical [3] ERP studies and MEG [4]. More recently, a consensus document has been published for both EEG/MEG studies [5]. A primer on MEG-EEG [6] is also available for students, postdoctoral fellows and faculty embarking on EEG/MEG studies. 

With improvements and miniaturization in technology, the number of measurement sensors in EEG and MEG systems has progressively increased. Data analysis methods, including artifact detection and removal strategies, have also been accordingly improved to accommodate data from these high-density recordings and to extract all information present in the EEG and MEG signals. MEG data were traditionally analyzed in source space, i.e., in terms of the estimated spatial distribution and time courses of the actual brain activity, while EEG more typically in sensor space, studying signals in the electrodes of interest. For the former, putative current sources in the brain were calculated based on observed activity at MEG sensors and a model of the head: the procedure is known as source modeling or source estimation, the latter emphasizing the fact that the signals of interest are obscured by noise. In addition, we prefer to reserve the term source modeling to the model of the elementary current source model, which is almost invariably a current dipole [7,8]. Source estimation has been said to be simpler to implement for MEG, unlike the electric potentials measured in EEG, as magnetic fields are largely unaffected by the electrical conductivity details within the head, such as the poorly conducting skull overlaid by a scalp with a better conductivity (but see [9]). For cases where the activity of interest is local, the spherically symmetric conductor model is sufficient [10]. The next level of complexity assumes that the skull is an isolator and the brain has a homogeneous, unknown, conductivity. This homogeneous head model [10] is usually sufficient for MEG modeling. Furthermore, it has been actually shown that MEG source estimates can be robust against errors in conductivity values when three compartments (scalp, skull, brain) are employed [9]. Nevertheless, the use of a three-compartment model has been recently recommended even for MEG [11], especially since it enables joint analysis of MEG and EEG using the same forward model. While accurate source estimation is possible based on EEG data, as well, the required head model is much more complex and must consider tissue conductivities and the individual shape of the compartments with different electrical conductivity, e.g., the scalp, skull, grey and white matter, as well as the cerebrospinal fluid [12,13].

In this century, EEG/MEG studies have begun to focus more on brain oscillations (rhythms) that are not strictly phase-locked to a stimulus or a motor action, or that occur spontaneously. In contrast, to evoked activity discussed above, stimulus-related activity lacking precise time or phase locking to the stimulus is commonly known as induced activity. These processes cannot be visualized by direct averaging, as their relationship to stimulus delivery is not constant. Therefore, any task can elicit both evoked and induced activity, which together are known as total activity. Additionally, there is considerable interest in the characteristics of brain’s spontaneous or “resting-state” activity, especially to determine the overall characteristics of brain connectivity. The study of social interactions between individuals has prompted the coupling of multiple EEG/MEG systems to one another; the method is often called hyperscanning [14]. Developments in technology are now also allowing portable dry-electrode EEG [15,16,17,18] to be used in naturalistic environments. In all of these cases, it is critical to preserve the integrity and quality of the data despite the considerable increase in experimental complexity.

Here, we briefly discuss issues and pitfalls that may be encountered while acquiring and analyzing EEG/MEG data while conducting studies in cognitive and social neuroscience. Practical considerations for performing different types of EEG and MEG studies are outlined. Our particular focus is on methods to minimizing artifacts during data acquisition and artifact removal during data analysis. We have drawn on material in earlier publications [1,2,3,4,5,6], as well as on our own experiences in using EEG/MEG.

## 2. Idiosyncrasies of EEG and MEG Studies

MEG requires a very specialized test environment that will allow the brain’s tiny magnetic fields, of the order of 10^−14^ to 10^−12^ Tesla, to be detected. MEG uses a purpose-built magnetically-shielded room (MSR) that filters out the Earth’s naturally-occurring magnetic field (10^−4^ T), and also fields generated by equipment (10^−7^ T), hardware such as elevators in the building, air-conditioning systems, or even trains and trams on the street outside. Devices that are used in the shielded room must be MEG-compatible, e.g., fiber optic response boxes, etc. If this practice is not followed, unwanted (non-brain) signals may be introduced by the device and cause artifacts in the MEG recording. For MEG the problem is more serious than for functional magnetic resonance imaging (fMRI) because even minuscule amounts of magnetic material not harmful in MRI can cause artifacts by far exceeding the level of the brain signals of interest. Therefore, MRI-compatible devices may not always be suitable for use in the MEG environment. For examples of MEG-compatible devices and stimulators, see Chapter 5 in [6].

EEG recordings have also typically been performed inside a shield, which is not as elaborate as that for MEG, because the shielding is for electric rather than magnetic fields. That said, however, the shielding will not be able to protect from inductive effects due to devices such as monitors, transformers, etc. Acceptable recordings can be obtained in a regular test room, provided that the room has been set up for this purpose. It is also important to make sure that the impedances of EEG electrodes are low and in the range recommended by the manufacturer and that the subject is not electrically connected to other devices. Sometimes, this is impossible. For example, EEG recording environments such as intensive care units and the operating room are very prone to artifacts, as the patient may be connected to multiple life-support devices. Fiber optic response boxes are recommended for EEG, as well, since they do not introduce artifacts into the recording, as could a regular button box or keyboard (see Figure 8.18 in [6]).

## 3. Study Design

### 3.1. What Is and Is Not Possible?

Traditionally, EEG/MEG studies have used tasks with unisensory stimuli, e.g., images, brief sounds, or electrical stimuli to a peripheral limb. Subjects had to remain as still as possible and keep eye blinks to a minimum. The same would also apply to resting state spontaneous EEG/MEG recordings, where subjects would be seated and instructed to keep their face and body movements and blinks to a minimum. Sometimes, subject movement can be an experimental confound, for example when it might systematically differ between different subject groups (patients versus controls, or young versus elderly subjects).

### 3.2. Naturalistic Stimulation

Most of the literature to date on naturalistic stimulation has used fMRI as an investigative tool.However, the sluggish nature of the hemodynamic response, which is indirectly related to actual brain activity, does not lend itself readily to imaging rapid changes in the environment. The excellent temporal resolution of EEG/MEG is ideally suited for these types of studies [19,20], and it is expected that in the future, EEG/MEG will be used more frequently for studies using naturalistic stimulation.

Existing studies of EEG/MEG in cognitive and social (neuro)science have been criticized for being artificial and lacking ecological validity. Our real everyday environment is dynamic and multisensory, both in terms of the presence of others, as well as from the point of view of changing environments, e.g., going from the indoors of one’s home to a car outside to drive to work, etc. More and more studies in cognitive and social neuroscience use complex stimulation, often in the form of movies with sound. The data have typically been analyzed with, e.g., correlational methods and reveal complex processes involving multiple brain networks that appear to be reproducible across individuals [21,22,23]. 

Relative to MEG, EEG has the additional advantage of being portable. The newest EEG recording systems with dry electrodes are portable and have Bluetooth/Wi-Fi capability [24,25,26] and can be controlled by a smartphone [27], so that subjects could actually take a walk outside the laboratory [28]. This development in the field is a very new one, and to date, there are only a few studies that compare new portable dry electrode and traditional fixed EEG systems with wet electrodes [25,28]. New sensor technologies, e.g., optically-pumped magnetometers operating at room temperature [29], open the possibility of producing small adjustable MEG sensors whose arrays could be moved to fit different head shapes, as well as having subjects potentially move around during a recording. However, eliminating artifacts due to movement of the sensors with the head in an inhomogeneous ambient magnetic field still remains a challenge, so recordings would still have to be performed in a shielded chamber.

For these naturalistic studies, a system of coding and tagging relevant events into the EEG data file as time stamps is crucial [30]. In naturalistic studies, using portable cameras to record a subject’s eye movements, as well as capturing video of the surrounding scene might be useful not only for data analysis, but also for the identification of unwanted non-brain signals or artifacts. This can be especially useful for special populations, such as infants and children, elderly subjects and neuropsychiatric patients. Additional time stamps to the EEG file can be added on post hoc inspection of the video and EEG file; however, this requires exact synchronization between the EEG system and the camera and some coupling of data files.

### 3.3. Hyperscanning

Interactive EEG/MEG studies with multiple subjects performing tasks are also possible; this requires two or more EEG or MEG systems to be coupled together. This has enabled social processes such as imitation of other’s movements [31,32,33], deployment of social attention [34] and verbal communication between mother and child [35] to be studied. The brain dynamics of four players of a card game has also been investigated [36].

Hyperscanning has multiple technological challenges [37]. First, multiple recording systems (and stimulus delivery systems, if appropriate) must be synchronized. Second, the amount of data acquired and stored is large. Third, data analysis and interpretation, typically examining synchronization of brain rhythms between subjects, requires specialized and purpose-written software. Fourth, similar brain rhythms can occur independently in subjects irrespective of whether an interaction task is being used. There is therefore a risk in these studies that spurious “hyperconnections” could be described [38]. 

Hyperscanning is becoming easier to perform with improvements in technology, especially using EEG recordings; however, methods development needs to progress further to deal with some of the measurement challenges, such as dealing with hyperconnections.

### 3.4. Concurrent EEG-MEG Studies

Commercial MEG systems usually include integrated amplifiers to record EEG and other electrophysiological signals. In these systems, the magnetic and electric signals are sampled simultaneously and the band pass filtering can be set to be identical. To avoid artifacts in MEG due to simultaneous EEG recording, nonmagnetic electrode caps, electrodes and leads should be employed as recommended by the manufacturer of the system. 

While EEG and MEG are both primarily sensitive to cortical currents, each method sees a slightly different aspect of the brain’s activity as each has a different sensitivity to the orientation and location of the cortical currents. Concurrent recordings of EEG-MEG can therefore generate a more complete picture of the profile of neurophysiological activity at a given point in time, and this is particularly important if an investigator wishes to estimate the active neural sources for a particular task or process. Importantly, however, for either EEG or MEG, strong activity from shallow sources (on the surface cortical mantle) can overshadow activity from that coming from deeper sources such as the insula, amygdala, etc.

MEG is sensitive to currents that are tangential to the surface of the scalp, as radial currents do not produce any magnetic fields outside the head. Since the net current direction on the cortex is perpendicular to the cortical mantle, this means that MEG is selectively sensitive to activity on the walls of sulci and is almost blind to that at the crests of gyri and bottoms of sulci. Activity from deep sources is also difficult to image, due to the large distance between the MEG sensor array and the neural sources themselves. MEG signals are not distorted by tissues such as the skull and the scalp.

In contrast, the EEG tends to be more sensitive to activity in the gyri, i.e., radial sources. That said, however, EEG can detect activity in sulci, so it is sensitive to both radial and tangential sources, provided that they are strong enough to be seen at the scalp. The EEG is therefore a more complex signal than that seen in MEG and can more readily record potentials from deep sources. However, the signal-to-noise ratio for deep sources may still be very low, as simultaneous superficial cortical activity usually produces much stronger EEG signals than distant deeper structures. In addition, the small size of the deep structures results in weaker effective currents compared to those on the large cortical mantle, and thus, the signal-to-noise ratio is further compromised. In recordings of spontaneous activity, the signal-to-noise ratio for deep sources may still be poor because of the overwhelming amount of simultaneously occurring activity at close-by gyri on surface cortex. Unlike MEG, EEG signals are also smeared and attenuated by tissues such as the skull and the scalp. Hence, for source estimation, the spatial resolution of EEG is poorer than that of MEG, but this may in part be due to inadequate modeling of the tissues of the head [39]. 

An important variable that can influence the accuracy of source estimation is that of the layout of sensor positions. Typically in MEG, sensors are arranged in an array with equal distances between sensors. Traditionally, in EEG studies, electrodes have been sited according to the 10-20 and 10-10 systems, where electrode positions are dictated by (uneven) 10% and 20% distances between the nasion and inion, as well as between the two inter-auricular points [40,41,42]. Most current simultaneous EEG/MEG studies have used these EEG electrode placement systems. High-density geodesic EEG electrode nets of 64, 128 and 256 electrodes [43] are now used more routinely used in EEG and have equal distances between sensors. These electrode nets tend to have quite high vertical profiles, creating a problem for simultaneous MEG, as the MEG sensors are located even further away from the scalp. To compare the relative contributions of simultaneously measured EEG and MEG activity in a balanced manner, one would have to use geodesic sensor arrays for both assessment modalities. Based on the existing literature, it does not appear that this has not yet been performed to date. 

The different sensitivities of EEG and MEG for detecting different types of neural activity suggest that simultaneous EEG/MEG studies can provide a more complete characterization of the brain activity pattern, particularly when source estimation procedures are used. The MEG data could be used to estimate tangential sources and, then, model the residual activity from radial and deep sources in the EEG dataset. An alternative method would be to combine the two datasets, but apply a weighting factor that is proportional to the signal-to-noise of each dataset [44]. It is known that fusion of EEG and MEG data can produce more accurate source estimates, as shown in studies with simulated and real data [39,44,45,46,47]. For example, it has long been proposed from EEG studies that an ERP occurring at around 100 ms after making an error (called the error-related negativity (ERN)) has an active source in the dorsal anterior cingulate cortex (dACC), matching the dACC focus of activation observed in fMRI studies (see [48] for a summary of studies). However, concurrent EEG-MEG recordings have indicated an origin for ERN in the posterior cingulate cortex (PCC). The fMRI Blood-Oxygen-Level Dependent (BOLD) signal in both dACC and PCC is correlated with ERN amplitude. Additional calculations of functional connectivity from the fMRI data indicate that the PCC generates the ERN and communicates with the dACC to enable error processing [48]. The PCC as a source of error-related neural activity has been corroborated in invasive recordings from monkey PCC during error commission [49]. 

One application where simultaneous EEG/MEG data acquisition is crucial is in the assessment of patients with focal epilepsies for potential seizure surgery, where the origin of the seizure activity must be reliably detected. Fusion of EEG-MEG data has been found to provide more complete information when identifying the location of an epileptic seizure [46], as well as inter-ictal epileptic discharges [50]. In particular, this is the case for detecting inter-ictal spikes where signal–to-noise ratios are typically quite low, as well as later propagated activity at subsequent time points following the spike [51]. Indeed, inter-ictal spikes will often manifest only in MEG or EEG, but not both assessment modalities, and this depends on the orientation, location and strength of the spike. Not surprisingly, EEG can detect spikes from deep sources, as well as have higher signal-to-noise ratios for radial orientations relative to MEG, whereas MEG was more likely to capture spikes with predominantly tangential orientations [52]. These data indicate how important the combination of EEG and MEG data are for scientific questions related to sources of activation (see Section 6.5).

One important question regarding detection of seizure activity and the potential excision of epileptogenic tissue is whether or not that tissue is functionally viable. Bast and colleagues (2007) studied 25 patients with seizure foci impinging or involving somatosensory cortex. Some patients had lesions as identified on structural MRI. In recordings of simultaneously-recorded somatosensory evoked potentials and fields in the normal and the affected hemispheres, they showed that viable evoked activity could be recorded from within the visible lesion and/or epileptogenic focus in the somatosensory cortex in five individuals. In some cases activity in MEG was not seen where EEG activity was present, and in one case, MEG activity was clearly seen, whereas EEG activity was not [53]. These data indicate how important the concurrent EEG-MEG studies can be in the clinical environment.

In the cognitive and social neuroscience sphere, concurrent EEG-MEG studies have not been so common and have been used to test or validate a new method of analyzing data. For example, an empirically-based Bayesian scheme for generating forward models (or the spread of activity through the head relative to measurement sensor positions, see Section 6.5, Source Estimation) has been studied using combined EEG-MEG data. Signal-to-noise ratios for each method were estimated and weightings of errors for each modality were factored into the automatic estimation procedure, which examined source estimation in face-evoked responses [54]. While localization of face-responses was reasonable with either EEG or MEG alone, the added benefit from combining the two datasets was demonstrated. More recently, brain activity to naturalistic viewing while EEG and MEG activity was recorded concurrently has been reported as a group of 12 subjects viewed a prolonged sequence from the movie “Crash” [19]. The investigators examined inter-subject correlations of activity as a function of viewed content calculated via a number of methods. Source estimation demonstrated consistent activations in occipitotemporal and superior temporal cortices that were correlated across subjects. The greatest synchronization in EEG-MEG activity occurred in the theta (6–8 Hz) range and to a lesser extent in the alpha band (10–12 Hz), likely reflecting the profile generated by evoked activity. The neural activity was calculated over sliding windows of 200 ms. This study is important, as it has demonstrated that recording and locating neurophysiological activity for naturalistic tasks is feasible. This approach has the advantage of being able to characterize the dynamics of the activation at a much finer time scale relative to an fMRI study and will probably be seen more commonly in the future.

### 3.5. Differences between EEG/MEG and fMRI Signals

Differences in activation loci in invasive neurophysiological and fMRI investigations have been reported [55], and it is expected that differences in scalp EEG and MEG studies would also be expected when compared to the fMRI BOLD response. This could be due to a number of reasons that we discuss below. Fundamentally, the nature of the neurophysiological and hemodynamic responses is very different: the former is a direct measure, whereas the latter an indirect measure of neural activity.

One major difference between the assessment modalities is that the temporal responses of the two signals differ by several orders of magnitude. EEG/MEG changes occur within milliseconds, while the hemodynamic response typically reaches its peak only after 4–6 s. It is therefore difficult to attribute specific neural activity that occurs in a discrete window of time in an experimental trial with the hemodynamic response, which likely reflects aggregated neural activity over a prolonged period of time. For example, infrequent, but strong, transient electrophysiological activity may be clearly detectable in EEG/MEG, while the corresponding hemodynamic change may be only minor. Conversely, weak, but long duration electrophysiological activity may produce a detectable fMRI signal that may be buried in simultaneous ongoing EEG/MEG.

A second major difference relates to the relative spatial sensitivities of each measurement method. In EEG/MEG, extended areas of active cortex could potentially be cancelled out if the activity happens to be located on opposite banks of a sulcus [56], producing no identifiable signal, whereas clear BOLD activation might be present. Conversely, in fMRI, the effects of partial voluming in voxels might dilute the signal if activity is confined to one bank of a sulcus, whereas it might be clearly seen in an EEG/MEG recording provided the source is strong enough and close enough to sensors.

A third major difference between assessment modalities could come about due to subject positioning during the experiment. Most EEG/MEG studies are performed with a seated subject, whereas a supine subject is studied with fMRI. Recent work has suggested that the profile of neurophysiological activity can be altered depending on whether the subject is seated or is lying down. Specifically, delta activity was increased for supine subjects irrespective of time spent lying down. Alpha activity was also decreased when the subject lay supine, but only when that occurred for a more prolonged (2 h) period of time [57]. In another study examining resting-state source modeled MEG data, recordings performed during sitting upright (versus lying supine) had greater widespread occipitoparietal high frequency activity (i.e., beta and gamma). In contrast, lying versus sitting upright produced more slower frequency activity mainly over the frontal cortex [58]. Near infrared spectroscopy (NIRS) signals are also altered as a function of body position (tilt), and this occurs for both oxygenated and deoxygenated signals [59]. It is therefore likely that BOLD signal changes might also be different depending on whether the subject is in a supine position or lies with their head and body on the side. This could be due in part on the pooling of blood in the posterior aspect of the brain when the subject is supine. These data have implications for comparison studies across the two modalities when performed in different sessions.

### 3.6. Concurrent EEG-fMRI Studies

Concurrent EEG-fMRI studies can capitalize on the superior spatial resolution of fMRI [60]. These studies are technically challenging because they require considerable neurophysiological expertise and fMRI-compatible EEG amplifiers that do not generate hazardous current flows that can cause burns to the subject and can also tolerate the large artifacts that are generated during MRI scanning. Typically, pre-processing of these EEG data must first remove the MRI scanning artifacts before tackling other physiological artifacts in the EEG signal that can be exaggerated in the MRI environment [61]. Concurrent EEG-fMRI studies are optimal for scientific questions relating to memory, language and priming, where recording data across two different sessions might produce different findings due to priming, practice or memory effects.

For EEG/fMRI data obtained concurrently, subject position is constant across assessment modalities, eliminating some of the issues discussed in the previous section. Current state-of-the-art EEG/fMRI studies are beginning to examine the relationship between brain oscillations and the hemodynamic response. High-field strength laminar fMRI indicates that EEG power in the different frequency ranges is related to BOLD effects in different cortical layers. Specifically, gamma band activity correlates positively with BOLD signal in superficial cortical layers, whereas negative correlations are seen for beta activity with BOLD in deep layers and for alpha activity in both deep and superficial layer BOLD [62].

In studies performed using EEG-fMRI at regular field strengths, i.e., 1.5–3 Tesla, the relationship between EEG/MEG and fMRI signals is a complex one. A good case in point is the observation that the power of the ongoing posterior alpha rhythm can be both positively and negatively correlated with BOLD response, with a critical issue being the amount of alpha power present at the time when a visual stimulus is delivered. Importantly, in trials that had high pre-stimulus interval alpha power, a nonlinear reduction of visual cortical BOLD activation and an associated increased suppression in auditory cortical and default-mode network (DMN) BOLD were observed [63]. These data indicate how important the effects of changeable activity in the pre-stimulus period can be, and have implications for differential measurements of the BOLD response, as well as for neurophysiological activity. Additionally, in resting-state experiments having the eyes open or closed makes a huge difference on the relationship between EEG activity and fMRI activation: ongoing alpha power is positively correlated with DMN BOLD activity when the eyes are open, and this is not the case when the eyes are closed [64]. 

Modulations in the power of EEG-MEG rhythms can be ultra-slow, i.e., activity <1 Hz or as low as 0.1 Hz [65,66,67], and it has been argued that these slow modulations may likely reflect a direct correspondence between EEG and fMRI activity [68]. The effects of the slow modulation of posterior alpha power have been found to extend to areas of brain well beyond visual cortex. Over a 20-min recording period, fast (0.04–0.167 Hz) and slow (0–0.04 Hz) fluctuations in posterior alpha power were measured. Slow fluctuations were positively correlated with BOLD changes in the brain stem, medial thalamus and anterior cingulate cortex, whereas fast fluctuations were correlated with BOLD changes in the lateral thalamus and anterior cingulate cortex, but not the brain stem. These data suggest that subcortical and cortical brain regions may modulate alpha power [69]. In future studies, it would be important to record neurophysiological activity invasively in all of these structures in addition to BOLD responses to more precisely characterize the direction and nature of this modulatory relationship. BOLD signals have also been found to vary up to frequencies of 0.75 Hz, and it has been argued that even faster changes in the BOLD signal may be possible [70].

It has long been proposed that the brain has continuously changing microstates whose duration can vary between tens to several hundreds of milliseconds. These microstates are identified by calculating global field power (GFP, across all EEG electrodes), and global dissimilarity (GD), defined as the standard deviation of the scalp voltage topographic map. The brain is said to transition from one microstate to another when GFP shows a minimum and GD a corresponding maximum [71,72]. Changes in brain microstates have been related to fMRI BOLD signal changes in resting-state networks [73,74]. It is clear from these early studies that there is much work to do in characterizing the relationships between EEG/MEG phenomena, BOLD signals and cerebral metabolism.

In principle, fMRI-guided Transcranial Magnetic Stimulation (TMS) with simultaneous EEG would be advantageous for identifying the critical time periods for performing various elements of a task in particular brain regions. For this to be effective, subjects would have to be studied with multiple modalities. Their previously-determined fMRI activation loci could serve as sites for single-pulse TMS at different post-sensory stimulus presentation times, determined by previous studies of evoked EEG/MEG activity. The combination of these neuroimaging modalities with TMS exploits the respective strengths of the three methods. Very few studies have attempted this, because this requires both varied technical expertise in the research group and a commitment from the subject to perform multiple parts of a scientific investigation.

### 3.7. Planning the Study

Irrespective of the study design used, experimenters should plan both data acquisition and analysis procedures completely at single subject and group levels before starting the study. Some analysis procedures require the data to be in a certain format or may require pre-stimulus baselines to be of a particular length relative to the total epoch length. These analysis considerations can potentially change how stimuli actually need to be delivered (e.g., inter-stimulus intervals), or how difficult the activation task might need to be, e.g., to ensure that there are enough trials and signal-to-noise in correct behavioral trials. Therefore, the hypotheses to be tested and, accordingly, the overall data analysis plan should be formulated a priori before actually collecting the data. These considerations do not preclude additional analyses being performed to follow-up unexpected patterns in the acquired data: hypothesis-driven and exploratory approaches can thus coexist harmoniously. 

It is also important to always include behavioral measures to provide context and inform the interpretation of neural measures. Depending on the study question, implicit measures of behavior may be appropriate, e.g., autonomic system variables such as changes in pupil dilation or heart rate variability.

## 4. Testing the Experimental Design Prior to Data Acquisition

One way to ensure that the timing of stimuli and events are recorded accurately during data acquisition is by performing a “dry run” of the experiment before acquiring real data (Figure 1). Correct scoring of behavioral responses can be tested by deliberately generating different types of errors. The experimenter can run the whole experiment and perform the behavioral task while recording dummy data so that subject responses will be time-stamped in the EEG/MEG data file. This will allow the data file to be scrutinized for correct timing of stimulus delivery, behavioral responses or other events related to the experimental protocol. This is especially critical if other devices are connected to the neurophysiological data acquisition system. For example, when performing EEG studies with TMS or fMRI, trigger pulses indicating TMS delivery or fMRI volume acquisition should be stored in the data file.

The test data should be analyzed using the full single-subject analysis pipeline. This procedure will allow any anomalies in experimental timing or issues related to the analysis pipeline itself to be identified and rectified. Following the correction of any errors in the data acquisition or analysis protocol, the dry run should be repeated until all errors have been eliminated prior to commencing actual data acquisition with subjects. Before starting to test subjects it is also advisable to run the experiment using a lab member as a test subject to iron out any additional issues in the protocol and data analysis pipeline from the real EEG/MEG data provided by the test subject. In addition, the group analysis procedures can be debugged and tested after data from a subset of the total subjects have been collected. A flowchart for testing and running a study is shown in Figure 1.

Hardware and software upgrades can sometimes generate surprises: an existing working data acquisition or analysis protocol can suddenly become inoperational following an upgrade. Worse still, an unknown modification may occur whose consequences are not immediately obvious. For these reasons, hardware or software is best updated after a protocol has been completed. It is also important to report the version numbers of the used hardware/software in manuscripts.

There will be obvious exceptions such as longitudinal studies, or multi-site studies, where specialized calibration procedures should be implemented for this contingency. For example, a phantom, which creates MEG/EEG signals for known current sources, can be employed to test hardware, as well as the data acquisition and analysis protocols. Additional testing can be performed with selected individuals who will perform multiple studies with a particular protocol [75,76]. 

## 5. Data Acquisition

### 5.1. Common Types of Artifacts in EEG/MEG Recordings

Unwanted non-brain signals or artifacts fall into two main categories: physiological and non-physiological. Physiological artifacts are electric and magnetic fields generated by, e.g., the heart, the muscles, particularly those of the eyes and face and the retina. These artifacts can be several orders of magnitude larger than the EEG/MEG signals of interest, but typically have a characteristic topography and can be dampened/removed by spatial artifact rejection techniques (as discussed in Section 6). For example, the effects of eye blinks and eye movements are seen in MEG and EEG sensors that are located on the face and over the anterolateral scalp (Figure 2). These artifacts can be minimized by having subjects fixate centrally and blink at the end of trials, but this may not be possible in naturalistic experimental designs. Blinks and eye movements can be easily monitored with electro-oculographic (EOG) measurements. That said, blinks and eye movements can potentially be problematic if they are time-locked to stimulus delivery or to behavioral responses. In contrast, electrocardiographic (ECG) activity tends to be largest on EEG/MEG sensors that are posterior and inferior and are most prominent in EEG recordings where sensors extend down onto the left side of the neck (Figure 3). This activity cannot be eliminated at data acquisition and requires artifact rejection methods to remove it during data pre-processing. For optimal cardiographic rejection, it is advisable to place ECG electrodes on the chest and include their output with the data. In contrast, sweat gland activity can be minimized by keeping the laboratory cool, but comfortable. Sweat glands produce potentials that will manifest as very slow, large amplitude excursions in the EEG. Additionally, sweating can change the impedance of the EEG electrodes and the skin, resulting in large changes in EEG baseline [77,78]. Should a subject begin to sweat, they can be cooled down with an electric fan. Muscle activity in the face and neck can be highly variable: it can involve different muscles whose spectral profiles can have quite different peak frequencies whose lower end can extend into the beta and even alpha EEG bands [79]; it may involve isolated muscles units or the whole muscle (Figure 4); and it can be present or absent at different times during the experiment (Figure 5).

Non-physiological artifacts include power line noise at 50 or 60 Hz (depending on the country) generated by any nearby electric equipment (Figure 6). Line noise can be a problem in both the EEG and MEG laboratory. For MEG, vibration can also be a problem. Nearby elevators, as well as trains/trams running by the building in which the MEG system is housed can create low-frequency artifacts in MEG signals. Similarly, air conditioning can also produce artifacts that have characteristic frequencies.

### 5.2. Reducing Artifacts during Data Acquisition

Experimenters should aim to record as artifact-free data as possible, as it is easier to prevent artifacts from occurring relative to removing artifacts from data post hoc. That said, for some naturalistic designs that we discussed earlier, artifacts may be an integral part of the dataset, since the scientific question being studied involves subject movement. In these studies, measuring EMG activity from selected muscles will be required to not only help code movements, but aid in artifact identification. Videoing the experiment could also be helpful. During the experiment, investigators should be monitoring EEG/MEG signals on-line through all sensors. Sometimes, artifacts can be spotted early in the experiment and eliminated if the experimenter talks with the subject during a regular break during a study if running a laboratory-based study.

Artifacts such as muscle activity, which can be particularly prevalent in tense and anxious subjects, can be reduced considerably if the laboratory environment is made to be as non-intimidating as possible. To this end, a comfortable subject chair, esthetic décor, low light levels in the lab and a pleasant and engaged experimenter can help an anxious subject relax. On the other hand, some subjects may be drowsy and in cases such as this an experimenter who is energetic can help subjects maintain a higher level of alertness for the study. Dividing the experiment up into runs of 4–5-min durations can also be helpful. Subjects can move and adjust their position so that they can stay comfortable for the duration of the study and hence minimize movement artifacts.

For experiments where subjects must maintain fixation and also minimize blinking, experimental designs that allow subjects to blink at the end of a trial are often used. In this way, blink artifacts during trials can be minimized. Some subjects can blink more frequently than normal during one of these studies because maintaining fixation and refraining from blinking can dry out the eyes. Similarly, if subjects normally wear contact lenses for vision correction, excessive blinking may occur; having subjects wear eyeglasses when coming to the lab may reduce this problem. Having some saline eye drops available in the lab can also be helpful. Note that having subjects refraining from blinking during experimental trials can itself be regarded as a task: the effects of this additional task on the experiment might need to be considered.

### 5.3. Additional Issues with Head Movements during MEG Studies

In MEG, the relative position of the head and the sensor array is not fixed. The head position during the measurement is typically monitored by recording magnetic fields from head position indicator (HPI) coils fixed on the head, so that the head position can be correctly taken into account in the analysis. That said, head movements should be kept to a minimum. Some MEG software packages provide a way to visualize the head position in real time after an HPI measurement is made, which can be very helpful in confirming that the subject’s is properly positioned with respect to the sensor array. During the off-line analysis, the head position information can be employed in various ways. For example, the signal-space separation (SSS) method [80,81] allows remapping the data into a desired head position. This makes it possible to conduct the subsequent source estimation and other analyses as if the head had been stationary during the measurements. However, it must be kept in mind that large head movements are still difficult to compensate for, especially because they may indicate that the subject is not performing the cognitive task or paying attention to the stimuli as requested.

Recently, some investigators have developed a cast for keeping the head still during MEG studies. Casts for individual subjects use the scalp surface obtained from their structural MRI, as well as the inner surface of the dewar and are constructed from foam resin. With the cast in place in the dewar, the head cannot easily move, so there is a potential risk of neck injury should the position of the dewar be altered while the subject is wearing the cast in situ [82].

### 5.4. Appropriate Filtering and Sampling Rates

The continuous EEG and MEG signals are digitized for subsequent storage. The Nyquist sampling criterion states that the sampling rate must be at least twice the highest frequency present in the input signal (see Figure 7). Therefore, prior to digitization, signals must always be filtered. EEG and MEG systems typically allow specification of a bandpass filter whose high pass and low pass limits, e.g., 0.1 and 200 Hz, should be selected to include all EEG/MEG signal frequencies of interest. Since the filter cutoff is not ideal, a generous sampling rate exceeding the Nyquist criterion should be selected. For example, one MEG manufacturer recommends that the sampling rate should be at least three-times the specified low pass filter cutoff. We recommend consulting the documentation provided with the EEG and MEG systems to decide appropriate filter and sampling settings. In addition to a bandpass filter, notch filters rejecting the line frequency and its harmonics are often available in the hardware. The acquisition-time bandpass should be kept wide because the data can be easily filtered to a specific band during off-line analysis.

We have already noted that inappropriate filter choices can introduce distortions (in the form of aliasing) and ringing in the data. It should be noted that analog filters (in the front-end of EEG/MEG systems) and digital filters (used in post-processing data) both suffer from aliasing and ringing. Additionally, both analog and digital filters can introduce delays or advances in EEG/MEG signals that are dependent on the cut-off frequency, which manifest as phase/latency shifts in the data. Delays or advances in phase/latency depend on the filter type: phase lags or slowed latencies can occur with low-pass filtering, whereas phase leads or faster latencies can occur with high-pass filtering [83]. Most of these effects can be avoided by using post hoc digital filters that do not induce phase shifts; however, it is advisable to use them on test data, to make sure that the filtered signals do not seem to move either forward or backward in time, which can be a problem for peak latencies for ERP deflections, as well as coherence and time-frequency analyses of EEG/MEG data [6]. Widmann et al. (2015) have provided an excellent discussion on the pitfalls of using filters in neurophysiology [84].

### 5.5. EEG Reference Electrodes

It is not unusual to use one reference electrode for data acquisition and then digitally re-reference the data during analysis. However, “linked” references such as the linked earlobe or mastoid configuration should not be used for data acquisition. This approach distorts the distribution of potentials across the scalp, and can produce false lateralization of activity [85] and the original data cannot be recovered in subsequent analysis. However, if the data need to be examined using this reference, digital re-referencing can easily be performed post hoc [6].

In principle, any single reference electrode can be used to acquire data, but thought should be given to what reference can minimize potential artifacts and also allow maximize visualization of experimental effects, particularly if a preliminary data analysis is being performed on-line during data acquisition. Common reference electrodes have included the nose, either ear/mastoid or the vertex. The nose has traditionally been a common reference for visual studies, but is prone to artifacts from the face, including blinks and eye movements. Currently, the vertex is often used as a reference and is probably one of the best choices for minimizing facial and neck muscle artifacts. In theory, the reference electrode should be located in a place that is inactive with respect to picking up brain activity. However, there is no place on the body that is inactive for measuring electrophysiological activity from either the brain or heart. The interested reader can find a summary of the rich discussion of the reference electrode issue in Hari and Puce (2017) [6].

It should be noted that some issues with reference electrodes can be encountered during data analysis (discussed in Section 6.3), but can be circumvented by working in source space (see Section 6.4).

### 5.6. Sensitivities of Different Types of MEG Sensors

MEG signals are measurements of the minute magnetic fields due to neural currents. These tiny fields are detected with pick-up coils connected to Superconducting QUantum Interference Device (SQUID) sensors. Two types of pick-up coils are generally used: magnetometers and gradiometers. The former measure some component of the magnetic field directly, while gradiometers are differential measurements providing rejection against far-away noise sources whose fields are spatially homogeneous. In an axial gradiometer, there is one coil loop close to the head and another, wound in an opposite direction, further away. The second loop detects only a minor contribution from the brain sources and thus the spatial patterns measured by axial gradiometers resemble those obtained with simple magnetometers. On the other hand, in a planar gradiometer the two loops are next to each other close to the head: a difference is obtained in the tangential rather than in the radial direction. The spatial pattern obtained with an array of planar gradiometers is thus different from that of a magnetometer or axial gradiometer array. 

Planar gradiometers pick up the largest signals if they are sited directly above the signal source, whereas axial gradiometer signals peak at two maxima of the dipolar field pattern [6], away from the source. In addition, two orthogonal planar gradiometers and a magnetometer at the same location measure independent information in the sense that their sensitivity patterns (lead fields) are orthogonal.

## 6. Preprocessing and Analysis

One of the first steps in the preliminary analysis of data is to identify and potentially remove artifacts in EEG/MEG signals. More traditional approaches include rejecting trials with artifacts outright; however, this requires designing an experiment with additional trials to allow for this contingency. Most investigators today use some form of artifact removal method. Below, we first briefly describe some artifact removal approaches that are specific for MEG followed by methods that can be used in both MEG and EEG.

### 6.1. Specialized MEG Artifact Removal

Two recently-formulated closely-related approaches to remove artifacts from MEG signals are known as signal-space projection (SSP) and signal-space separation (SSS). Both methods remove a spatial subspace due to noise sources from the data. In SSP, the noise subspace is determined experimentally by recording data with subject absent [86]. In SSS, the noise subspace is modeled using a physics-based multipole expansion of sources outside the sensor array: the noise should be contained in this subspace. In addition, SSS employs a source model to reconstruct the measurements, as they would have appeared in the absence of the disturbance and its removal [80,81]. A similar reconstruction procedure can be applied after SSP as well [87], but is generally not necessary because the same effect can be achieved by taking SSP into account in the forward modeling of sources [88]. It is also worth noting that especially SSP can be employed to dampen blink and eye movement as well as cardiac artifacts. For this purpose, the empty room noise subspace is augmented with additional projection vectors determined from periods of data containing this artifact in the absence of task-related activity.

### 6.2. Removing Artifacts in MEG or EEG Data with Independent Component Analysis

One commonly-used approach is to remove artifacts using an independent component analysis (ICA), which exploits the fact that different artifacts can have stereotypical spatial patterns and temporal signatures. ICA is a “blind” source separation method meaning that it is completely data-driven, and attempts to separate the dataset into different sources that are independent in both space and time [89]. The sources could be artifacts or brain signals, e.g., alpha activity etc. While many investigators use ICA for removing artifacts from EEG/MEG data, some groups also use ICA for extracting wanted signal features in EEG/MEG [90]. Semi-automated approaches that remove artifacts under experimenter supervision are available [91]. Artifacts such as eye-related activity such as blinks have a characteristic anterior distribution, making it easy to identify and remove from the EEG/MEG data (see Figure 2). Eye movement activity, e.g., lateral gaze shifts in a central fixation task with lateral stimulus presentations can sometimes be visualized, and can also serve as indicators of subject compliance with experimenter instructions. Similarly, ECG-related activity can be effectively removed using ICA because of its characteristic temporal pattern and spatial distribution that is typically seen in the left lower posterior scalp, particularly in electrodes that cover the neck (Figure 3a). On rare occasions ECG-related activity can be split across two ICAs, as shown in Figure 3b.

Line noise in EEG data will depend on electrode impedances that can progressively change during a long experiment (Figure 6). ICA components showing prominent line noise typically will have bizarre topographies that reflect the relative impedances of the electrodes: electrodes with highest impedances relative to the reference will show the largest line noise artifacts. If the recording session is long and EEG electrodes begin to dry out, then line noise might also progressively increase as a function of time (see Figure 6b).

Similarly, muscle activity can be highly variable in terms of its frequency, as well as spatial distribution during the course of an experiment (Figure 4 and Figure 5), so that it is very challenging to remove with any sort of analysis. It is important to note that EMG activity can even appear in the beta and alpha ranges, and the frequency is dependent on the muscle that generates the activity [79]. Some investigators also use ICA-based methods to identify non-artifactual features in the EEG, such as task-related activity [92]. For a detailed discussion on artifact removal using ICA see Hari and Puce (2017) [6].

Artifact rejection methods such as signal-space projection (SSP) [86], SSS and ICA modify the MEG/EEG expression of the actual brain activity as well. Therefore, the effect should be taken into account in source estimation correctly by modifying the forward model. Most academic software packages can do this in a correct, well-documented manner, but this may need to be explicitly specified by the user.

### 6.3. Digital Re-Referencing of EEG Data

As we noted earlier, the choice of reference electrode, which is an EEG-specific issue, may differ between data acquisition and analysis. What should dictate the choice of reference electrodes for data analysis? An excellent reason for selecting a particular reference electrode might be to try to link current findings to previous literature. Some of the existing EEG literature may seem conflicting and this may be due in part to sensor-of-interest based peak amplitude measures of event-related responses. In these cases quite different measurements of amplitude can be obtained (e.g., see Figure 5.6 in [6]). This is not a problem when working in source space or when examining scalp topographic maps in sensor space where the topography itself is not influenced by the reference electrode. If a sensor-of interest analysis is necessary, then it is suggested that investigators draw on the previous literature and use electrode clusters that have been used in previous studies. If reference electrodes have differed across previous studies, it may be useful to express the data as a function of different reference electrodes, for the purposes of trying to explain some of the discrepancies in the previous literature, e.g., [93,94]. Another reason to re-reference EEG data post-hoc might be that particular analysis packages may require data to be expressed with respect to a particular reference, e.g., average reference for some source estimation packages [95,96,97]. Reference electrodes can be problematic for some measures of functional connectivity as some analyses may yield artificially inflated values of the measure of interest, e.g., coherence and the average reference is a known issue [98]. Ideally, analyses of functional connectivity should be performed in source space. This may not be possible, however, for some recordings, e.g., low-density EEG electrode arrays that are typical of portable EEG systems. In these cases, either the data using the average reference should be expressed as Laplacians (second spatial derivative of the EEG data) or re-referenced to a single reference electrode [98,99,100].

There has been a long-standing discussion regarding the ideal reference electrode for data analysis, and while there is no general consensus between investigators, some points need to be considered. While the average reference electrode is commonly used, it should be noted that there are issues with its use. The assumption behind its use is that provided that there is spatial coverage across the entire head and measurement sensors are numerous and equidistant, the sum of the potentials across the head at a given point in time should always be zero [101]. This assumption can be violated and ghost potentials can be introduced [102], particularly in recordings with small numbers of sensors and uneven spatial coverage. However, the average reference has been noted to be suitable for high-density EEG recordings (of 128 channels or more) [103,104,105]. As already noted, the average reference should not be used for analyses of coherence, as it can artificially inflate coherence values [98,99,100].

An “infinity” reference has been touted as being one of the best references to use [106] and is appropriate for measures of coherence [107]. The infinity reference expresses EEG data with respect to a point located at infinity relative to the brain and the measurement sensors, and hence requires data to be analyzed in source space (see below) because it is reconstructed from solving the forward problem [108]. 

Examining the data with respect to different reference electrodes can link the current data to various older studies in the literature, and investigators might consider publishing data with respect to more than one reference to make an explicit connection with the earlier literature [6,94].

It should be noted that the issues we highlighted with EEG reference electrodes above are not a consideration if the data are analyzed in source space.

### 6.4. Sensor and Source Space Analyses

As already mentioned in the introduction, sensor space data refers to EEG/MEG signals at each measurement sensor, i.e., in the form that they were acquired in. In contrast, source space data refers to the amplitudes of the estimated sources in the brain. Source space data is obtained as a result of source estimation, to be discussed below. From the outset, MEG data have been often expressed in source space, whereas EEG data are still more commonly analyzed in sensor space. As we noted earlier, this was due in part to the fact that MEG will detect activity from tangential sources, whereas EEG signal will be more complex as it will “see” mainly activity from radial sources and detect activity from tangential sources as well. 

One could argue that all analyses should be performed in source space, however, there may be good reasons for continuing to work in sensor space. For example, portable EEG systems typically have low-density sampling across the head, with too few sensors with which to perform source estimation reliably.

### 6.5. Source Estimation

The goal of source estimation is to identify the active neural sources that contribute to an observed EEG or MEG pattern in sensor space and requires solving the so-called inverse problem. The inputs to a source estimation procedure are the EEG/MEG data themselves, information about the locations and orientations (for MEG) of the sensors with respect to the head, and an adequate forward model, which predicts the MEG/EEG signals due to sources in the brain and allows comparison of the data actually measured and those given by a set of candidate sources. The source estimation procedure adjusts the locations and orientations of the putative sources until a best match is obtained. In addition, a noise estimate, usually a noise covariance matrix, is employed to regularize the solution: some mismatch between the measured and predicted data is tolerated to make the estimates less sensitive to noise. Since the electromagnetic inverse problem does not possess a unique solution, additional constraints are required to render the solution unique. The art of MEG/EEG source estimation is to select these constraints appropriately and to interpret the resulting estimates correctly. 

One common approach is to assume that the cortical activity underlying the measurements is sparse, i.e., salient activity occurs only in a small number of cortical sites, and that each active area has a small enough spatial extent to be equivalently accounted for by a point source, a current dipole. This multidipole model has been developed to great sophistication [109] and has been very successful in the analysis of event-related potentials and fields. Figure 8 displays an example of multidipole modeling of neural responses in the somatosensory system to electrical stimulation of the median nerve at the right wrist in a single subject [110]. In this experiment, the inter-stimulus interval was variable (mean 2.3 s, range 1.5–21 s) and the subject was asked to respond to each stimulus with the index finger of the left hand. Approximately 120 epochs were averaged. A multidipole analysis revealed four sources: one at SI in the left hemisphere, followed by activity in secondary somatosensory cortices (SII), best explained by one dipole in each hemisphere, in the inferior parietal lobe near the insula. Finally, a motor response associated by the button press was observed in the primary motor cortex (MI) in the right hemisphere. As shown in Figure 8, the responses in the somatosensory cortices were very well defined while the MI signal, averaged to the onset of the stimulus rather than the response is spread out in time. Figure 8 also demonstrates that if the dipole locations and orientations are assumed, it is possible to detect single-epoch source time courses as well. In addition to dipole time courses, the goodness-of-fit to the data should be evaluated as a function of time as well. This information is valuable, in particular, in interactive analysis to find the correct number of sources, see, e.g., [111]. Strictly speaking, the significance of the model should be determined with help of a statistical test, but in the case of uniform noise levels across sensors and a fixed sensor configuration the goodness-of-fit value is a useful surrogate.

A second approach assumes a distribution of sources on the cortex and to apply an additional criterion to select a particular distribution to explain the data and to produce an image of the most likely current distribution. To date, the most successful approach of this kind is the cortically constrained minimum-norm estimate (MNE) [87,112,113], which constrains the currents to the cortical mantle and selects a solution, which has the minimum overall power. An example of using distributed sources in the analysis of visual data without fMRI constraints is shown in Figure 9 [114]. In this case, activity in the visual system has been estimated to visual stimulation using the face-vase illusion, where different parts of the image, i.e., the faces and the vase, were subjected to dynamic noise at different frequencies (12 and 15 Hz) that can be clearly differentiated in an occipital MEG sensor as shown by power spectrum and time-frequency analysis plots. Importantly, when the subject’s perception switched from “faces” to “vase”, there was an associated change in the respective frequency of MEG activity. The sources responsive to the change of the percept are best seen in the medial views of each inflated cortical hemispheric surface. This example indicates that frequency tagging can be an effective stimulation tool for differentiating activation in distributed source models where the anatomical loci are likely to exhibit spatial overlap.

Additional constraints in the form of fMRI activations can also be applied to these types of model, e.g., [115], as the example in Figure 10 demonstrates. Here distributed sources have been modeled in supratemporal cortex to selective attention to auditory identity (phoneme) or location. In a paired stimulation paradigm an adapter preceded a probe stimulus by 3.4 s. The adapter and probe could be the same or different with respect to location or identity. Source activity in supratemporal cortex peaking at around 100 ms was extensive and reproducible to the adapter (left panels). Activity to the probe at the same latency varied in three ways: a. when adapter and probe were identical there was widespread suppression of the response across supratemporal cortex (bottom row); b. when the sound location changed between the adapter and the cue, activity in the anterior aspect of auditory cortex was suppressed (white arrow, top panel); c. when sound identity was changed, suppression in activity in the posterior aspect of auditory cortex was observed (white arrow, middle panel). This example indicates how relatively subtle activation patterns can be identified with the distributed source approach.

A third approach of source estimation involves dipole scanning, where a suitable scanning function, derived from the input data, is evaluated at each candidate source location. A high value indicates a likely location of a source. Examples of these type of methods are the linearly-constrained minimum-variance beamformer [116,117] and multiple signal classification (MUSIC) [118,119] approaches. The beamformer method has gained a lot of popularity among MEG researchers while its use in EEG has been limited. This is probably due to the fact that for the beamformer method to work, the forward model needs to be sufficiently accurate [120]. This issue is complicated because the required accuracy of the forward model depends on the signal-to-noise ratio of the data: for low-noise measurements, a more accurate forward model is necessary. Successful use of the beamformer technique even for MEG, therefore, usually calls for regularization of the data covariance matrix, which effectively artificially increases the noise level [121]. The scanning approaches differ from the parametric dipole model and the source imaging approaches in the sense that the ‘pseudo-images’ they produce do not constitute a current distribution which is capable of explaining the measured data. However, the methods are still closely related because the scanning approaches are evaluated on the basis of their ability to correctly locate the active sources. On the other hand, the time course output, e.g., by a multidipole model, can be considered as outputs of a spatial filter, which makes linear combinations of the signals in the original measurement channel. A recent publication [122] presents a general framework for generating purpose-built filters on the basis of existing knowledge about the sources to yield high-fidelity source time courses.

Since anatomical information is needed both to construct an accurate forward model and for constraining the locations and orientations of the putative sources, it is advisable to obtain a whole head structural MRI scan for each subject. Source estimation and anatomical modeling packages often make explicit recommendations on the particular types of MRIs to be acquired for best results.

## 7. Keeping Records

### 7.1. Cataloguing and Characterizing the Data

For rigor and reproducibility, it is imperative that adequate records are kept in the laboratory that properly catalog the characteristics of the raw and analyzed data, so that they can be accessed for many years in the lab by individuals who might not have performed the experiments. Given that analysis of neurophysiological data has become so complex, ideally a digital record of the analysis pipeline and its details should be generated from the analysis of each single subject’s data, as well as any group or cumulative analysis. Similarly, details and specifics of the statistical analysis of data should also be adequately documented so that it can be understood by individuals who did not actually collect or analyze the data. This also includes details of simulated data, or data distributions that have been made for statistical comparisons such as permutation analyses. It should be possible to re-generate the identical simulated data if needed. Some academic MEG/EEG source modeling packages have means to effectively document the analysis pipelines [95,105,123] and thus provide an excellent starting point to promote reproducible research and to share methods across laboratories. Furthermore, the Brain Imaging Data Structure (BIDS) standard is being extended to MEG/EEG [124] and will provide a means by which to access data easily from different software packages.

### 7.2. Data Storage and Accessibility

The data should be available to all individuals who work in the laboratory, or at least be accessible with established procedures, e.g., accessing it from data archives, to enable future comparisons of newer studies to older data.

## 8. Data Sharing

The culture of the practice of science has progressively changed, with increasingly more open sharing/access of source code and data [125,126,127]. Databases and associated meta-analyses for fMRI-related data have existed for some time, but have not typically included options to include EEG/MEG data. The NeuroVault database has been designed to be a repository for the extraction of statistical maps, typically from fMRI data, but does include EEG/MEG data as well, provided that these have been expressed in source space [128]. 

Due to the varied nature of data analyses in the EEG/MEG area and their expression in either sensor or source space, EEG/MEG databasing is associated with considerable challenges. Indeed, meta-analyses of ERP data are few, partly because of the variability in data formats and the way in which studies have been described in the literature. Below we describe some data sharing initiatives that will provide meta-analytic and training opportunities for EEG/MEG researchers.

An attempt to create an ERP-based database has been made by a consortium of researchers who have established the Neural ElectroMagnetic Ontologies (NEMO) database [129]. Its aim is to give researchers data that are in a common format with specified information with respect to how the data were acquired and analyzed. A set of guidelines detailing the information required for submission to the database are available [129].

With respect to MEG-related data, the Open MEG Archive (OMEGA) is an open-neuroscience initiative, which has been initiated to curate both raw and processed data for open dissemination with the scientific community [126]. While its primary focus is MEG, other associated data such as structural MRI, subject questionnaires and simultaneously acquired EEG data can also be included. There are also websites that point to available shared EEG/MEG datasets such as [130]. Another large openly available data set from The Cambridge Centre for Ageing and Neuroscience (Cam-CAN, [131]) provides epidemiological, behavioral, and neuroimaging data for the purpose of understanding how individuals can best retain cognitive abilities into old age.

Additionally, large-scale initiatives such as the Human Connectome Project (HCP) have been designed to acquire multimodal data from a large number of individuals with standard protocols in fMRI and MEG [132]. The HCP provides an opportunity for researchers all over the world to access and perform analyses on these large datasets, allowing the possibility of performing standard analyses with adequate power, as well as testing out new analysis methods.

Testing new methods of analysis often require both real and simulated data. To this end, MEG-SIM is a web portal with examples of simulated data for testing analysis methods [133]. This is an important resource, given that our analysis methods are becoming more and more complex, with errors in coding being increasingly difficult to detect.

The neuroimaging community has always been a very interactive one, and we expect continued discussions at scientific meetings, in journals and on social media with respect to issues and standards related to data sharing to continue in the future.

## 9. Special Considerations for Studying Clinical or Pediatric Populations

### 9.1. Subject Comfort and Optimal Data Acquisition

Recording from patients can be challenging: individuals who are in physical discomfort or who may be mentally impaired can find regular EEG or MEG studies challenging. Patients may have trouble sitting still and avoiding blinking. The same can be said for pediatric studies of EEG and MEG. Having toys on hand and poster/pictures on the walls for subjects (and their siblings) can make the experience for the subjects and their families more pleasant. The potential benefit can be a considerable improvement in the quality of the collected EEG/MEG data.

### 9.2. Specialized Head Models for Pediatric Studies of EEG

The brains and skulls of infants and toddlers are in a state of constant change. At the age of one year an infant head is approximately 90% of the size of an adult head, reaching 95% by six years of age. Discontinuities in the skull (fontanels) progressively close and this process can continue until 24 months of age, which is followed by thickening of the skull [134]. This has tremendous implications for EEG recordings, as the spread of electrical activity can potentially be quite different relative to a skull that is closed and also has attained its full thickness. Therefore, structural MRI scans of the head should ideally be obtained of each subject, so that an accurate head model can be constructed should source modeling form part of the EEG data analysis plan. Given that obtaining a structural MRI scan of the head is often not possible with this subject population, some investigators choose to use head models that consist of an average of a group of healthy subjects. However, in the past some studies of infants have used the (closed skull) templates of older children for data analyses using source modeling. This may lead to localization errors, which are generally significantly smaller in MEG than in EEG [135]. Additionally, there is the potential to overestimate the strength of the active sources, if the fontanels are not taken into account in the forward model: the open (and thinner) skull will lead to larger EEG amplitudes than predicted by a model with a closed skull. 

Age appropriate head models in infants and children are beginning to be available to investigators, so if an individual MRI scan is not possible, an age-appropriate head/brain template can be used. For example, brain templates are available for children 2 weeks to 4 years of age [136]. As noted earlier, ideally the structural MRI scan should be of the head and not just the brain and be obtained from the same individual as close as possible to the time of the EEG recording. 

## 10. Conclusions: Final Remarks

The EEG/MEG community is growing, with investigators from many different disciplines joining the effort to study the brain. Analysis methods continue to improve, but also become increasingly complex. Our field faces a number of challenges with respect to training new researchers, as well as for providing opportunities for investigators to share data. It is important that we employ common terminology based on solid concepts for communicating our science. This includes reporting data acquisition and analysis procedures in a way that other researchers can reproduce the methods and can replicate the study. Here, we have provided suggestions and resources towards this end, as well as discussing some of the pitfalls and possible solutions. We hope that EEG/MEG will continue to grow as the methods of study in cognitive and social neuroscience.

## Figures and Tables

**Figure 1 brainsci-07-00058-f001:**
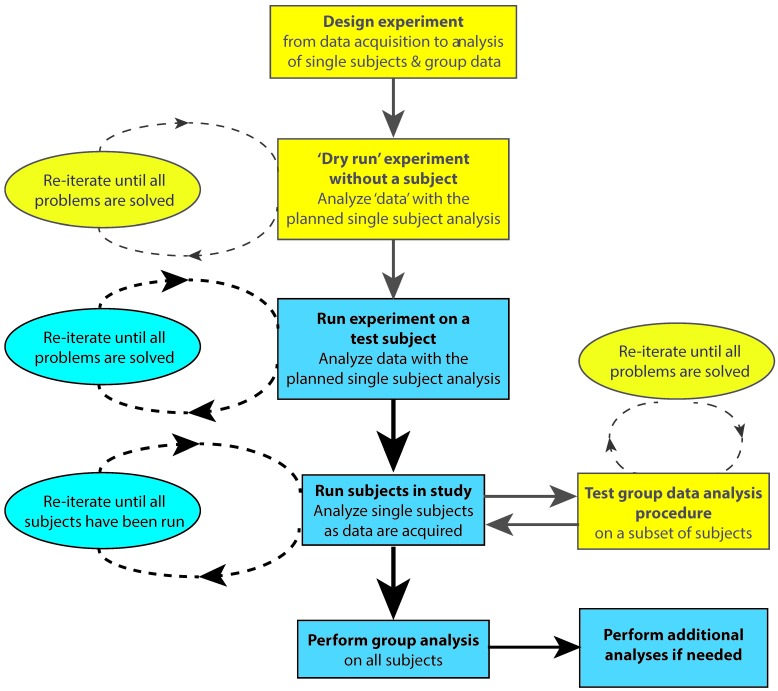
Flowchart for implementing a neurophysiological study. Various stages in planning and testing (yellow boxes) as well as executing (blue boxes) an experimental study are shown. See the text for further discussion.

**Figure 2 brainsci-07-00058-f002:**
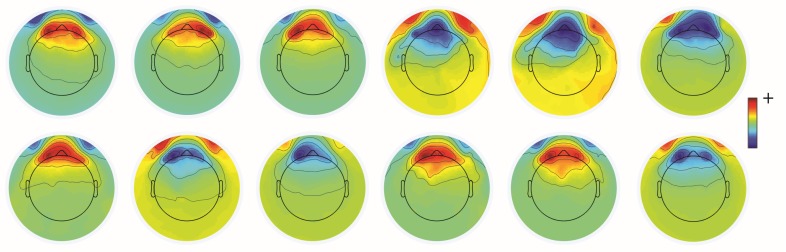
Topography of eye blinks. Maps from 12 individual subjects display the typical anterior topography of eye blinks as identified by independent component analysis (ICA) in EEG data (see also the text in the “Preprocessing and Analysis” section for more detail on the method). Nose is at top, inion at the bottom, and left and right hemispheres appear on the left and right. The color scale depicts arbitrary units.

**Figure 3 brainsci-07-00058-f003:**
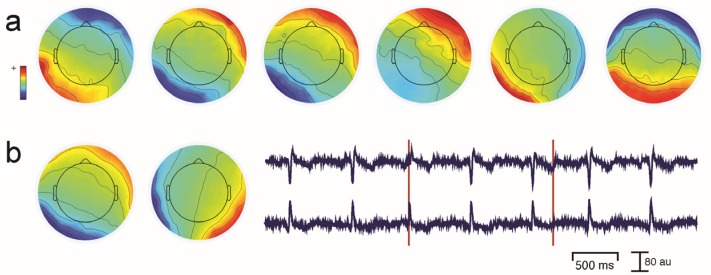
Topography and time course of the electrocardiographic (ECG) artifact. (**a**) Topographic maps (six subjects) showing the characteristic topography of ECG artifacts in EEG data as identified with independent component analysis (ICA) (see also the text in the “Preprocessing and Analysis” section for more detail on the method). A diagonal distribution, with a gradient from the left posterior scalp (where the ECG signal is typically maximal) to the right frontal scalp (e.g., see the first four subjects on the left) is seen. The exact artifact topography depends on the position of the cardiac source in the chest. Some individual variation in scalp ICA maps (e.g., see the last two subjects at the right) can be seen. The color scale depicts arbitrary units. (**b**) On rare occasions, the artifact appears in two separate ICA components, as shown in the data of one subject. The ICA maps show the diagonal gradient (left) and an additional component with a different gradient. The ICA time courses of the ECG artifacts (right) show clear QRS complexes in the epoched data traces (red vertical lines separate data epochs). a.u. = arbitrary units.

**Figure 4 brainsci-07-00058-f004:**
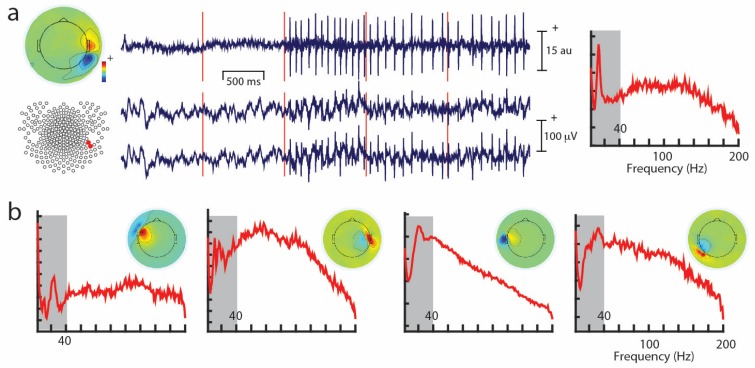
Muscle unit spiking artifact. (**a**) A topographic map from Independent Component Analysis (ICA, left panel, top) and the associated scalp EEG electrodes (bottom left, red circles) overlying a muscle unit. The muscle unit’s ICA time course (middle panel, Trace 1) and corresponding EEG data (Traces 2 and 3, with red vertical lines separating data epochs are shown). The spiking of the muscle unit appears in the third epoch and is seen above some residual muscle activity. The muscle unit’s spikes can be clearly seen in the EEG signal and coincide temporally with spikes in the ICA time courses. The associated logarithmic power spectrum (right panel) shows a prolonged tail of EMG activity out to 200 Hz, but also has a prominent peak in the EEG frequency range (0–40 Hz, gray shading) caused by the spiking of the muscle unit. a.u. = arbitrary units, µV = microvolts. (**b**) Examples of spatial distributions EMG ICA components with their associated power spectra. All show prolonged tails up to 200 Hz with different peaks, some of which clearly occur in the EEG/MEG frequency band (0–40-Hz range, gray shading) in all plots. Peaks in the alpha and beta ranges are caused by the spiking behavior of the muscle units.

**Figure 5 brainsci-07-00058-f005:**
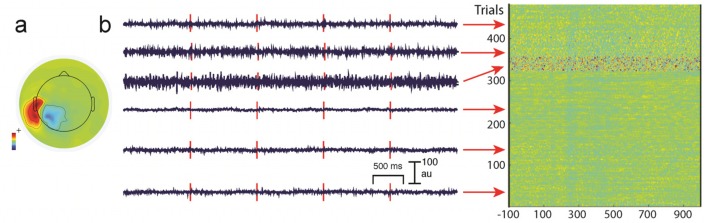
The varying nature of muscle activity. (**a**) Topography of muscle artifact, as identified with an independent component analysis (ICA) map from EEG data. The color scale depicts arbitrary units. (**b**) Six ICA time courses showing electromyographic (EMG) activity from five epoched single trials (red vertical lines) that waxes and wanes during the course of an experiment (bottom to top: increasing time during the experiment). a.u. = arbitrary units. Maximum EMG activity is seen in the third trace from the top. At the right, a compressed display of ICA time courses across the entire experiment (colors depict arbitrary units) of a total of nearly 500 trials is shown. The red arrows identify time points during the experiment to which the time courses correspond. The display also shows a period of pronounced EMG activity from around trials 325–330 at which the EMG activity was maximal, as also shown by the third time course at left.

**Figure 6 brainsci-07-00058-f006:**
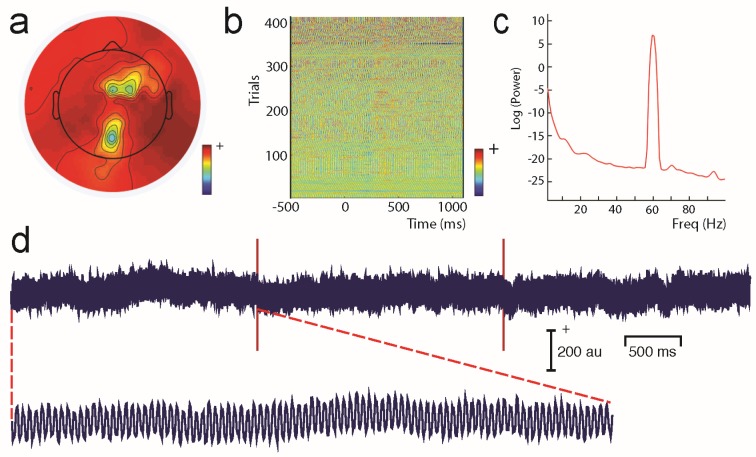
Line noise changes during an experiment. (**a**) Topographic distribution of 60-Hz line noise artifact in EEG data, as identified with an independent component analysis (ICA) map. The bizarre topography of the artifact reflects the likely difference in EEG electrode impedances across the head. The color scale depicts arbitrary units. (**b**) Compressed display of ICA time courses over the entire experiment shows an increase in the size of the artifact over time. Trials increment from bottom to top, and the artifact is more clearly seen as blue and red colors at the top of the display. The color scale depicts arbitrary units. (**c**) Power spectrum of the ICA component shows a clear peak at 60 Hz. (**d**) Time course of the ICA component across three epochs shows the “fuzzy” appearance of line noise that is typical in EEG recordings. The bottom trace shows an expanded version of the signal with a clear periodic form as a function of time.

**Figure 7 brainsci-07-00058-f007:**
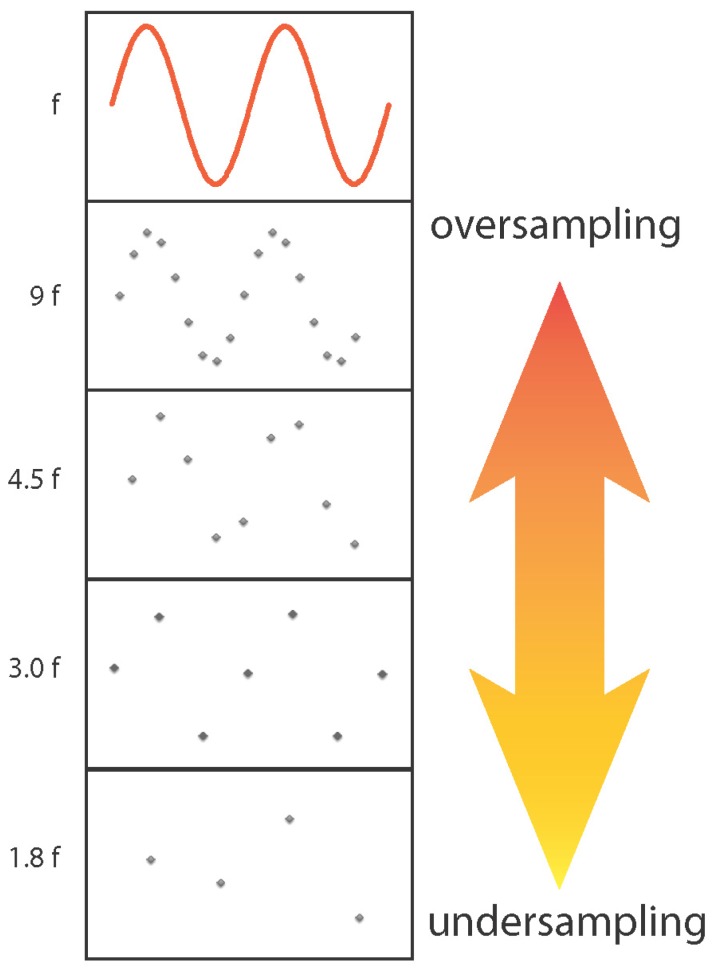
Effects of sampling rate on sampled data**.** A sinusoidal signal, f (top left), has been sampled at a number of different rates relative to its frequency. Progressively slower sampling reduces detail and signal quality. At frequencies above the Nyquist criterion of 2f, the frequency content of the signal can be seen (middle three traces). However, note that information has been lost in the bottom trace where the data have been sampled at 1.8f.

**Figure 8 brainsci-07-00058-f008:**
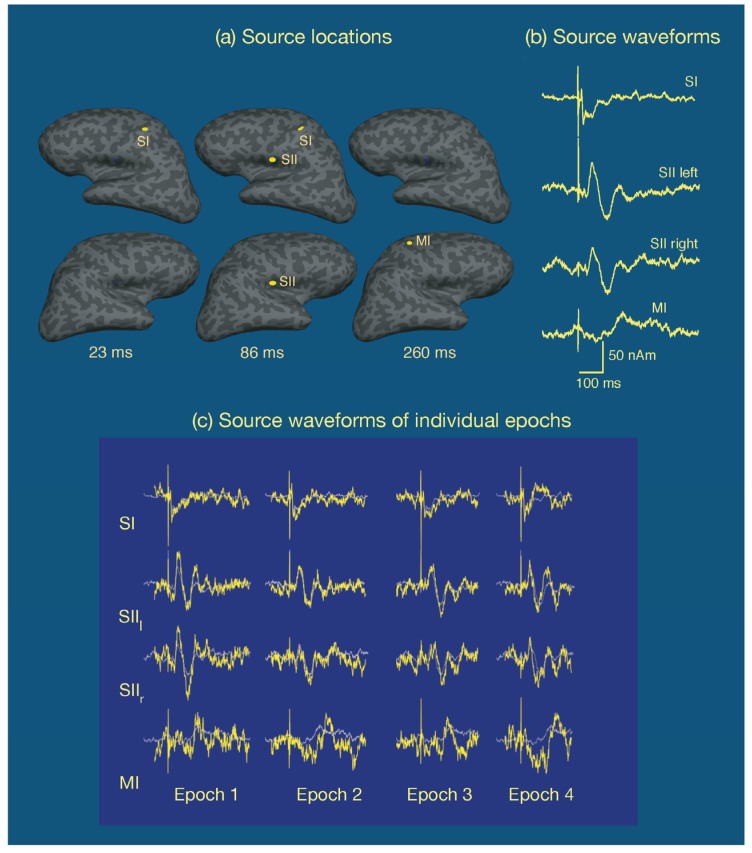
Multidipole model of activity in the somatosensory system. (**a**) Estimated locations of somatosensory and motor sources visualized on the inflated representation of the cortex at three time points during the epoch. (**b**) Estimated time courses of the four sources. The vertical line in each trace denotes time of stimulus delivery. (**c**) Estimated source time courses for four individual epochs assuming that the source locations and orientations are the same as those determined from the average responses. Light blue: the source time courses calculated from the average. Yellow: the source waveforms based on individual epochs.

**Figure 9 brainsci-07-00058-f009:**
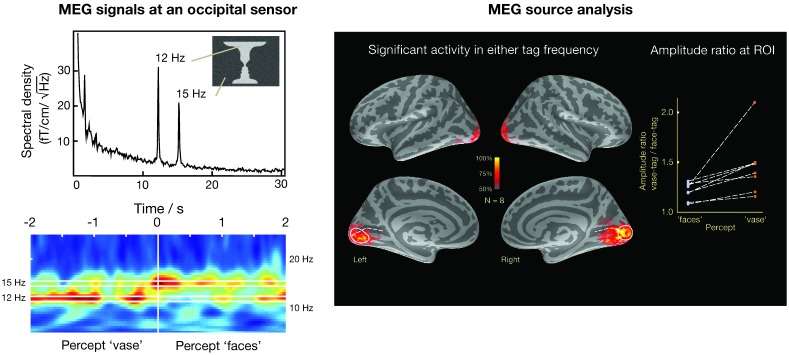
Distributed source estimates without functional magnetic resonance imaging (fMRI) constraints in the visual system. (Left) An indication of a percept is seen in early visual areas with help of frequency-tagged stimuli. Top plot shows a power spectrum with two clear peaks at each tagging frequency. Bottom plot depicts a time-frequency analysis with the white vertical line corresponding to the time at which there is a perceptual change. There is a clear shift in peak frequency of activity from one tagging frequency to the other (12 Hz–15 Hz) at this point in time. (Right) Change in the relative strength of activity in an occipital MEG sensor when the percept changes. Location and strength of activity is depicted on inflated lateral and medial brain surfaces in the left panel. White broken lines identify the calcarine sulcus, whereas white sold lines denote a region of interest. The right panel depicts amplitude ratios for stimuli at each tagging frequency. For details, see the text.

**Figure 10 brainsci-07-00058-f010:**
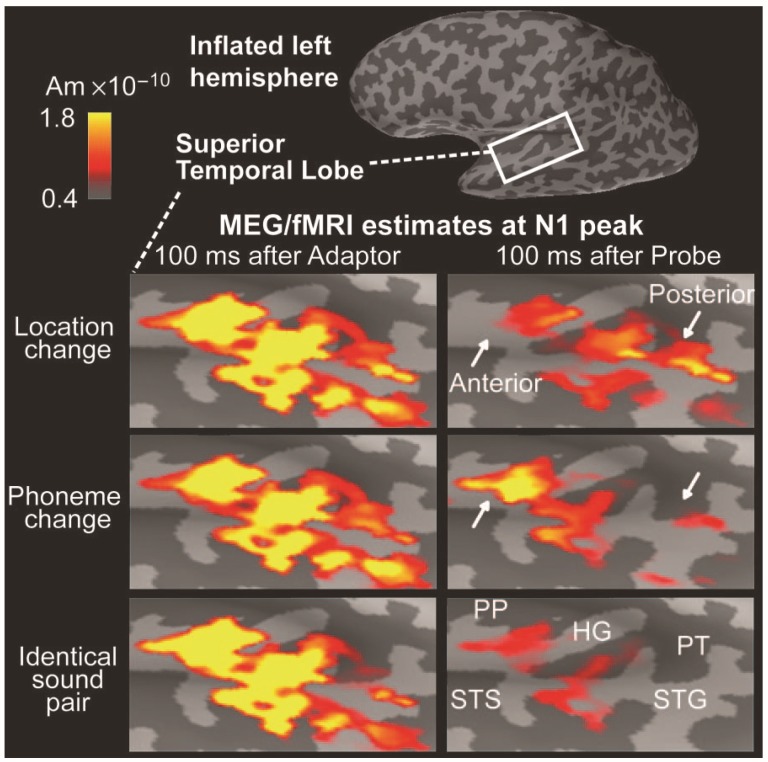
Task-modulated distributed source estimates in the auditory cortex. Distributed functional magnetic resonance imaging (fMRI)-weighted cortical source estimates at auditory cortex marked by the rectangle in the inflated cortical view (top). Source estimates depicted at 100 ms after the adaptor stimulus (left column) and after the probe stimulus (right column). The three rows show the different adaptor-probe combinations. See the text for details. PP = Planum Polare, HG = Heschl’s Gyrus, PT = Planum Temporale, STS = Superior Temporal Sulcus, STG = Superior Temporal Gyrus.

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
