# Peer review of "A Review of Issues Related to Data Acquisition and Analysis in EEG/MEG Studies"

_brainsci, 2017, doi:10.3390/brainsci7060058_

Round 1

Reviewer 1 Report

This paper presents a review on the different aspects of EEG/MEG data acquisition, analysis, storage and sharing, especially emphasizing on artifact removal or minimization. Although, there exists several papers providing guidelines of EEG or MEG practices, this paper could be considered as a reference for current and new investigators to adopt the standard EEG/MEG practices as it has reviewed on materials from all the published documents on EEG/MEG practices and provided the issues and pitfalls of EEG/MEG experiments. The paper is overall well written and contains interesting and relevant information. While the paper has been well-presented with sufficient references, examples and technical details, few corrections as indicated below should be considered before publication.

Line 17 in the Abstract à the correct adjective is de rigueur 

Line 40 à best practices has been published …

Line 70-71 à give a reference for the dry electrodes EEG system here.

Line 97 à may be connected to many other types…

Line 122 à remove the exclamation mark.

Line 132 à the word “yoking” does not seem like an appropriate word here, may be another word like “couple”?

Line 135 à same as line 132.

Section 3 and Section 3.6 have literally the same name. May be Section 3.6 should be named Data Analysis.

In Section 3.4, add references on difference between EEG and MEG sensitivities. When talking about the differential sensitivities of EEG and MEG about the orientation and location of the cortical currents, here it should be emphasized why simultaneous EEG and MEG can be helpful. Both EEG and MEG record neuronal activity coming from same cortical currents with high temporal resolution but provide information about different aspects of the cortical currents.

Plus, this Section 3.4 could be elaborated a bit more with few examples of simultaneous EEG-MEG studies. The next section on EEG-fMRI is more detailed and seems unfair to section 3.4. Since the Section 3.4 is under study design, it is important to talk about simultaneous EEG-MEG experiment design such as it entails MEG-compatible EEG system and amplifiers, then mention what are the existing MEG and EEG system that can allow simultaneous EEG-MEG acquisitions, also how this combined EEG-MEG data has been utilized so far (different integration strategies such as comparative integration, constrained integration, symmetrical integration). Some references for this would be:

1.      Bast T, Wright T, Boor R, Harting I, Feneberg R, Rupp A, Hoechstetter K, Rating D, Baumgärtner U. 2007. Combined EEG and MEG analysis of early somatosensory evoked activity in children and adolescents with focal epilepsies. Clinical Neurophysiology. 118:1721–1735.

2.      Kirsch HE, Mantle M, Nagarajan SS. 2007. Concordance between routine interictal magnetoencephalography and simultaneous scalp electroencephalography in a sample of patients with epilepsy. J Clin Neurophysiol. 24:215–231.

3.      Henson RN, Mouchlianitis E, Friston KJ. 2009. MEG and EEG data fusion: Simultaneous localisation of face-evoked responses. NeuroImage. 47:581–589.

4.      Fuchs M, Wagner M, Wischmann H-A, Köhler T, Theißen A, Drenckhahn R, Buchner H. 1998. Improving source reconstructions by combining bioelectric and biomagnetic data. Electroencephalography and Clinical Neurophysiology. 107:93–111.

5.      Ding L, Yuan H. 2013. Simultaneous EEG and MEG source reconstruction in sparse electromagnetic source imaging. Hum Brain Mapp. 34:775–795.

6.      Chowdhury RA, Zerouali Y, Hedrich T, Heers M, Kobayashi E, Lina J-M, Grova C. 2015. MEG–EEG Information Fusion and Electromagnetic Source Imaging: From Theory to Clinical Application in Epilepsy. Brain Topogr.:1–28

Line 154 à in spontaneous activity the signal-to-noise ratio may be still poor….

Line 194 à …whereas negative correlations ….

Line 225 à …the behavioral task while recording dummy…

Line 348 à …available in the lab can also be helpful.

Line 482 à add the reference for brainstorm software package.

Author Response

Please see the attached document - with responses to each of the Reviewer comments.

Reviewer 2 Report

This is a timely, useful and interesting review and overview of practical and theoretical issues associated with EEG/MEG recordings and analysis. In my view, both beginners and experts will be benefit from it. It complements and updates the existing guidelines and handbooks on EEG/MEG and ERP/ERF analysis. I still have a number of comments, mostly clarification issues, that will hopefully help to further improve the manuscript. 1) It might be useful to mention EEG/MEG already in the title. Section 1 ends with a statement that EEG/MEG analysis will be illustrated with examples from “social neuroscience”. However, the illustrations are not all from social neuroscience. If the authors indeed want to focus on this area (which I wouldn’t necessarily recommend because it would narrow down the readership), they should already highlight this in the abstract 2) l. 55: The authors state that MEG signals are “largely unaffected by the electrical conductivity details within the head”. This is true if the emphasis is on “conductivity”, but head geometry still matters, and realistic head models (at least with one compartment, or approximations) are advised. This could be clarified. A recent publication that has shown robustness of MNEs to conductivity errors is for example https://www.ncbi.nlm.nih.gov/pubmed/23639259. 3) ll. 92ff.: In my view, the benefits of Faraday cages for modern (low impedance) EEG recordings are often over-estimated. For example, it doesn’t shield from inductive effects (monitors, transformers etc.). It is more important to make sure the participant is not electrically connected to other devices, impedances are low, etc. 4) section 3.1: It could be mentioned here and elsewhere that body movements (limbs, head, eyes etc.) can be serious confounds between participant groups (e.g. patients and controls, old and young etc.). If possible, this should be quantified, e.g. by obtaining some movement parameters during the recordings as is standard in fMRI analysis. 5) L. 108: The point about ecological validity is true for most studies in cognitive (neuro)science. 6) L. 116: The “sluggish response” argument could already be made at the very beginning as a selling point for EEG/MEG. 7) L 120: Do you want to mention the development of new MEG sensor types (e.g. OPMs) that might in the future allow MEG recordings during (some) movement (also 5.6)? 8) 3.3: It should be pointed out that “hyperscanning” is currently much easier, and has already been done, with EEG. 9) L. 151: activity from deep sources may also be completely overshadowed by activity from more superficial sources, i.e. it’s also a problem of relative sensitivity. 10) L. 163: I think differences between fMRI and EEG/MEG are more common than similarities, especially for more complex cognitive functions. The physiology of the two modalities are fundamentally different. This would deserve a more detailed discussion, with appropriate references, in a separate section. 11) L. 186: The point about supine/seated position was already made before. 12) L. 189f: I can’t follow the argument about low frequencies. I think the best correspondence between BOLD and EEG is usually found in the gamma band (at least for simultaneous EEG/fMRI recordings)? 13) Figure 1: add “recurrent loops” to parts of the figure to indicate trouble-shooting and optimisation? (I think the second sentence of the caption would be nice with a verb) 14) Section 5.1: already refer to artefact correction methods here? 15) Figure 2: Which measurement modality is shown here (I assume EEG)? 16) Figure 3: Is that much detail on ECG necessary? Also, check “(a)” and “(b)” labelling in figure. 17) Figure 4: It would be helpful to see the frequency spectrum of muscle artefacts here. 18) L. 313: you could also mention air-conditioning here, which can have characteristic frequencies. 19) L. 335: “non-intimidating” or “comfortable” instead of “threatening”? 20) 5.2: The requirement to sit still is less severe for EEG compared to MEG. In MEG, one should make sure that one monitors head movement accurately, within and across sessions. If subjects move between sessions, they should be re-positioned as accurately as possible. Eye blink are particularly “dangerous” when they are time-locked to stimuli or responses. 21) 5.3: Different MEG systems deal with this differently (e.g. not all monitor continuous head movement). Maybe mention Maxfilter options here? 22) L. 422ff: You could mention that leadfields for magnetometers and planar gradiometers at one location are orthogonal, thus providing independent information, and most efficient sampling. 23) 6.2: You could mention that if one corrects data with ICA etc., one should also adjust the forward model with the corresponding linear transformation. 24) L. 484: I don’t think using multiple reference configurations is a very practical suggestion. It may encourage researchers to try several options and use the “best”. The choice of reference should be based on the research question at hand. 25) L. 494f: Functional connectivity analysis in sensor space is limited in several ways, but I don’t think the average reference is a particularly serious problem. Any reference will introduce spurious connectivity. 26) L. 505ff.: The definitions of sensor and source space should be provided when the terms are first used. 27) L. 511: this could already be mentioned above, when problems of reference electrodes are described. 28) Figure 8: Dipole time courses can be problematic, since the model may not fit well across the whole latency range. 29) L. 590f.: I agree with the description of beamforming as dipole scanning. Some authors highlight the interpretation of BFs as “spatial filters”, rather than as a localisation method. However, other methods have this property, too. A possible reference for this would be https://www.ncbi.nlm.nih.gov/pubmed/23616402. 30) L. 592: The discussion of the effect of head modelling errors (how big? How accurate does a head model have to be for certain applications?) would deserve more discussion in a separate section. 31) Figures 9 and 10: I would suggest swapping the sequence of these figures. It sounds like using fMRI constraints is the standard, but one can also do it without it. I would start with the more typical application. 32) L. 613: If you provide references for analysis pipelines, it should be complete rather than single out only a few examples. For example, SPM also allows scripting, as do at least some of the commercial packages. You could also mention the BIDS for MEG development: http://bids.neuroimaging.io/. 33) Section 8: Another big openly available data set would be http://www.cam-can.org/. 34) 9.1: this is mostly obvious and repetition from previous sections. It might be more important to point out that noise, movement etc. can be a serious confound for between-group studies. Typos etc.: Some more references could be provided for: l. 70: dry-electrode systems 192: laminar fMRI l. 40: “HAVE been published” 154: signal-to-noise RATIO 194: negatively 228ff: strange sentence l 231: I understand why ‘data’ is put in ‘’ – but would be better to give them a clear name, e.g. “test data”? 257: “more than an order of magnitude”? 286: “(“e.g. 531: “into” 536: Be consistent about hyphen in “multi-dipole” 583: DIPOLE scanning Olaf Hauk (in case clarifications needed)

Author Response

(The authors gave the same response as above.)

Round 2

Reviewer 2 Report

The authors have made substantial changes to the manuscript, and have addressed all my comments. I would still like to make a few minor points.

The authors say that average reference is the worst choice for coherence analysis. Every signal at the reference electrode will be subtracted into the signals of all other electrodes. If people choose (linked) mastoids, for example, they may spread auditory signals or even muscle artefacts across all electrodes. The average signal across an ideal EEG array (surface around head) would be zero. In my view, this is the least biased one among the practical choices.

l. 196: do you mean "moveable MEG sensors"? "Portable" might be confused with portable EEG systems, but MEG will still need a shielded chamber.

l. 228: What kind of "artefactual signals" are you referring to? It does not produce artefacts in the MEG. It may produce EEG artefacts if the participant moves, but this can also occur for EEG alone. The brain is a little bit further away from the MEG sensors, which in my view is compensated for by the gain in information from EEG.

l. 820: "IT should"

l. 895: I'm not sure how many software packages correct the forward model after ICA. The inverse operator doesn't necessarily "know" where the data come from, i.e. the experimenter has to explicitly inform the analysis.

Author Response

Please see the attach word doc for our comments to the Reviewer.
